# Cross-Sectional Anatomy and Computed Tomography of the Coelomic Cavity in Juvenile Atlantic Puffins (Aves, *Alcidae*, *Fratercula arctica*)

**DOI:** 10.3390/ani13182933

**Published:** 2023-09-15

**Authors:** José Raduan Jaber, Marcos Fumero-Hernández, Juan Alberto Corbera, Inmaculada Morales, Manuel Amador, Gregorio Ramírez Zarzosa, Mario Encinoso

**Affiliations:** 1Department of Morphology Faculty of Veterinary Medicine, University of Las Palmas de Gran Canaria, Trasmontaña, 35413 Las Palmas, Spain; 2Veterinary Clinical Hospital, Faculty of Veterinary Medicine, University of Las Palmas de Gran Canaria, Trasmontaña, 35413 Las Palmas, Spain; juan.corbera@ulpgc.es (J.A.C.); inmaculada.morales@ulpgc.es (I.M.); mencinoso@gmail.com (M.E.); 3Dirección Insular de Medio Ambiente, Consejeria de Área de Medio Ambiente, Clima, Energía y Conocimiento of the Cabildo de Gran Canaria, 35002 Las Palmas, Spain; mcamador@grancanaria.com; 4Department of Anatomy and Compared Pathological Anatomy, Veterinary Faculty, Campus de Espinardo, University of Murcia, 30100 Murcia, Spain; grzar@um.es

**Keywords:** coelomic cavity, computed tomography, anatomical sections, seabirds, puffin

## Abstract

**Simple Summary:**

Birds constitute an important group within the new companion animals. Since their popularity has been increasing, it is more common to find them in homes, zoos and wildlife centres around the world, evidencing the need for clinicians, biologists and researchers to have a deep knowledge of their anatomy, pathology and physiology to provide adequate medical care.

**Abstract:**

In birds, unlike mammals, there is no complete separation between the thoracic and abdominal cavities. Instead, they have the coelomic cavity where most main organs are found. Therefore, an adequate knowledge of the anatomy of the coelomic cavity is of great importance for veterinarians, biologists and the scientific community. This study aimed to evaluate the coelomic cavity anatomy in the Atlantic puffin (*Fratercula arctica*) using anatomical sections and computed tomography images.

## 1. Introduction

Birds constitute an enormously diverse taxonomic group, including thousands of species worldwide [1]. These animals have historically caused the fascination of humans, who have adopted them as companion animals. The possession of birds as pets is a growing trend, and their presence in zoos, wildlife and veterinary centres is also increasingly common [2]. For all these reasons, knowledge of the anatomy and pathophysiology of these animals has become one of the objectives of current veterinary medicine.

Imaging diagnosis is an essential tool for bird clinicians since many psittacine species exhibit a special ability to mask disease signs [3]. Conventional radiology is the method most widely used as an auxiliary diagnostic technique in avian medicine due to its low cost, non-invasive nature and the fact that it is found in most veterinary centres [4,5,6]. However, modern imaging techniques such as magnetic resonance (MR) and computed tomography (CT) make it possible to obtain three-dimensional images with little distortion and even simulations of dynamic processes of different organs and vascular structures in a minimally invasive way [7,8,9]. Consequently, its use in anatomical studies has spread, and even smaller versions, such as micro-CT, have been created for animal research [10,11,12,13,14,15,16,17,18,19,20,21]. These anatomical studies are essential not only from the clinical point of view but also for endangered species conservation, enabling adequate knowledge of their anatomical characteristics and behaviour patterns [21,22,23].

A diverse range of studies can be found regarding the use of diagnostic imaging in the study of the coelomic cavity of birds; however, as far as we know, there are no studies that deal with the anatomy of the avian coelom using the Atlantic puffin as a model [24,25,26,27]. The Atlantic puffin (*Fratercula arctica*) is a medium-sized marine bird of the Alcidae family with characteristic black and white colouration. It lives mainly in the North Atlantic Ocean, where it faces a multitude of threats, many of them linked to human activity. All this has led to it being classified as a vulnerable species on the list of threatened species of the International Union for Conservation of Nature (IUCN) [28,29].

The aim of this paper was to study the coelomic cavity of the Atlantic puffin using anatomical sections and CT images, which could serve as an anatomical model for pathologic studies of other phylogenetically related birds.

## 2. Materials and Methods

### 2.1. Animals

In the present study, a total of 20 juvenile Atlantic puffin (*Fratercula arctica*) carcasses were utilised. The avian specimens displayed an average mass of 0.251 kg (with a range of 0.185–0.310 kg) and an average length, measured from the beak tip to the base of the tail, of 19.85 cm (with a range of 16–24 cm). The study group comprised young animals that were provided by the Consejeria de Área de Medio Ambiente, Clima, Energía y Conocimiento of the Cabildo de Gran Canaria following a massive stranding event on the island’s coastline. Subsequently, all enrolled puffins for this study were sectioned and scanned at the Veterinary Hospital of Las Palmas de Gran Canaria University (Canary Islands, Spain).

### 2.2. CT Technique

For the CT examinations, the carcasses were thawed at room temperature for 12 h. Sequential transverse CT slices were acquired using a 16-slice helical CT scanner (Toshiba Astelion, Canon Medical System^®^, Tokyo, Japan). The animals were symmetrically positioned in dorsal recumbency on the stretcher, with craniocaudal entry. A standard clinical protocol was followed, employing the following parameters: 120 kVp, 80 mA, 512 × 512 acquisition matrix, 1809 × 858 field of view, pitch of 0.94 and a gantry rotation of 1.5. The obtained images had a slice thickness of 0.6 mm. To enhance the assessment of the coelomic structures on CT, three different CT Window settings were applied by adjusting the window widths (WWs) and window levels (WLs): a bone window setting (WW = 1500; WL = 300), a soft tissue window setting (WW = 248; WL = 123) and a lung window setting (WW = 1400; WL = −500). No significant variations in CT density or anatomy were detected within the coelomic cavity of the birds used in this investigation. Finally, all these images were uploaded to an image viewer (OsiriX MD, Apple, Cupertino, CA, USA) to facilitate data manipulation and analysis.

### 2.3. Anatomical Sections

Following the acquisition of CT images, the imaged cadavers were frozen at −80 °C for 72 h. Subsequently, eight specimens were sectioned in the transverse plane. To accomplish this, parallel sections, each one centimetre thick, were generated using an electric band saw. Finally, after carefully moistening the sections obtained with water, removing the artefacts (feathers and sand) with Adson forceps and identifying them, they were photographed on both sides.

For further examination and identification of internal structures in anatomical cross-sections and CT images, three puffins were dissected to expose the coelomic cavity and visualise the location of the organs. The dissection facilitated the accurate identification and correlation of anatomical features observed in anatomical sections and CT images.

### 2.4. Anatomic Identification

For the purpose of identifying and labelling the cross-section along with the corresponding CT images, we used reference materials, such as textbooks and articles from scientific journals focused on avian anatomy [30,31,32]. Furthermore, to ensure precise anatomical interpretation of the coelomic structures, we also used different anatomical preparations provided by the Department of Anatomy of the Faculty of Veterinary Medicine, University of Las Palmas de Gran Canaria. These additional resources proved beneficial in enhancing our understanding and accuracy in interpreting the anatomical features within the coelomic cavity.

## 3. Results

Different figures revealing the anatomical structures of the Atlantic puffin coelomic cavity are presented (Figure 1, Figure 2, Figure 3, Figure 4, Figure 5, Figure 6, Figure 7, Figure 8, Figure 9, Figure 10, Figure 11 and Figure 12). Figure 1 comprises a compilation of various dissections displaying the main structures within the coelomic cavity. Figure 2 shows a sagittal multiplanar reconstruction (MPR) volume rendering, wherein each line and corresponding number (I–IX) represents approximately the level of the following anatomical and CT transverse planes. Figure 3, Figure 4, Figure 5, Figure 6, Figure 7, Figure 8, Figure 9, Figure 10 and Figure 11 consist of four images for each case: (A) an anatomical cross-section, (B) a pulmonary CT window, (C) a soft tissue CT window and (D) a bone CT window. These images are presented in a rostrocaudal progression, starting from the lungs (Figure 3) to the cloaca levels (Figure 11). Finally, Figure 12 is composed of three images: (A) an anatomical dissection and (B,C) dorsal MPR volume rendering images in the pulmonary CT window at different levels.

### 3.1. Anatomical Dissections and Cross-Sections

We present anatomical dissections (Figure 1A–C and Figure 12A) and transverse (Figure 3A, Figure 4A, Figure 5A, Figure 6A, Figure 7A, Figure 8A, Figure 9A, Figure 10A, Figure 11A) cross-sections of the coelomic cavity. All these figures were essential in facilitating the identification of the organs of the respiratory, circulatory, digestive and urinary systems within this cavity. Consequently, we identified the puffin’s heart, which exhibits an oval shape with a sharply pointed apex located along the central axis of the coelomic cavity and cranial to the liver (Figure 1A–C). Moreover, the heart chambers, including the Right and Left Atrium and the right and left ventricles, were observed. Additionally, these images were crucial in visualising significant blood vessels, such as the Left and Right Brachiocephalic Trunks, the common carotid artery, the left cranial vena cava and the left jugular vein (Figure 1A,B). The right and left lungs were visually discernible in a craniodorsal location, ventral to the thoracic vertebra and lateral to the ribs. The anatomical dissections and transverse cross-sections also aided in distinguishing the trachea (Figure 1A,B). This structure coursed in a median position into the coelomic cavity until bifurcating into the right and left main bronchi (Figure 3A, Figure 4A, Figure 5A). In contrast, we could identify the syrinx in specific dissected images (Figure 1B and Figure 12A). Dorsal to the trachea, we visualised other formations, such as the oesophagus, the longissimus colli and the sternotracheal muscles (Figure 1A,B and Figure 12A).

The dissected and transverse images played a crucial role in observing the walls of the air sac. As a result, the cross-sections allowed the observation of the topographic distribution of the different air sacs, including the clavicular, the cervical, the thoracic and the abdominal air sacs (Figure 1A,B, Figure 3A, Figure 4A, Figure 5A, Figure 6A, Figure 7A, Figure 8A, Figure 9A, Figure 10A and Figure 12A). These anatomical preparations also facilitated the observation of the puffin’s liver, which consists of the right and the left hepatic lobes, whose cranioventral segments surrounded the heart (Figure 1A,C and Figure 12A). Both lobes exhibited similar sizes and were larger than other organs within the coelomic cavity. At the level of the visceral surface of the right lobe of the liver, a membranous structure corresponding to the gallbladder was distinguishable (Figure 12A). Moreover, the medial border of the hepatic lobes showed the hepatopericardial ligament, which connects the liver with the heart apex (Figure 1C). Caudally and closely related to the visceral surface of the left hepatic lobe, we identified the muscular portion of the puffin’s stomach, known as the *ventriculus* (Figure 1A,C and Figure 12A). Its identification was facilitated by the presence of a thick muscular layer, as well as by the visualisation of two muscles, the *crassus caudodorsalis* and the *crassus cranioventralis*, which delineate the dorsal and ventral borders of the *ventriculus* (Figure 8A and Figure 9A). In addition, these sections also allowed us to observe its glandular part, the *proventriculus*, which shared a similar location at the left side of the coelomic cavity after the junction with the oesophagus and the *ventriculus* (Figure 6A and Figure 7A). Other digestive system components, such as the small and large intestine, were also visible in these anatomic sections. The small intestine (*duodenum*) was better identified in the dissected image caudal to the liver (Figure 1A,C and Figure 12A). It showed a U-shaped *ansa duodeni* comprising a descending segment, *the pars descendens*, and an ascending portion, the *pars ascendens* (Figure 1C and Figure 12A). More caudally, we could distinguish the terminal intestine (*caecum*), which was well developed in the puffin, exhibiting a distinctive colouration (Figure 1C and Figure 12A). Between these two segments and close to the duodenal loops, we observed the pancreas that lies within the mesoduodenum, displaying a pale yellow colouration (Figure 1C, Figure 8A, Figure 10A and Figure 12A). In addition, the utilisation of transverse sections was crucial in discerning the presence of paired kidneys situated laterally to the spine and embedded dorsally in excavations of the synsacrum (Figure 8A, Figure 9A, Figure 10A). Interestingly, these transverse sections allowed us to visualise the ventral surface of the right and left kidneys in close contact with the paired abdominal air sacs (Figure 8A and Figure 9A). Furthermore, the caudal pole of the kidney housed the ureters, as depicted in Figure 9A and Figure 10A. These structures were distinguished emerging from the ventral aspect of the kidneys. Concerning the genital organs, we distinguished a round single yellow structure that closely resembled the ovary (Figure 6A).

In the caudal part of the coelomic cavity, we could identify an excretory passage for the digestive and urogenital systems known as the cloaca, which was surrounded by cloacae muscles (*musculus levator cloacae and musculus trans* versus *cloacae*) that allows the cloaca to become larger for different functions, including egg laying, copulation and defecation (Figure 11A). In addition, various bony structures related to the coelomic cavity were observed, including the vertebrae, ribs, sternum, thoracic and pelvic limbs, as well as the lateral parts of the pubis. These bone formations were associated with specific muscles, such as the different parts of the Pectoral Muscle (sternobrachialis, thoracic and abdominal portions), the Supracoracoid Muscle, the scapulohumeralis muscle, the scapulohumeral caudal muscle and the Longissimus Dorsi Muscle (Figure 3A, Figure 4A, Figure 5A, Figure 6A, Figure 7A, Figure 8A, Figure 9A, Figure 10A, Figure 11A, Figure 12A).

### 3.2. Computed Tomography Images

The CT images that closely corresponded to their anatomical sections were selected (Figure 3B–D, Figure 4B–D, Figure 5B–D, Figure 6B–D, Figure 7B–D, Figure 8B–D, Figure 9B–D, Figure 10B–D, Figure 11B–D and Figure 12B,C). These CT images provided additional morphological and tomographic details regarding the coelomic structures when compared with the corresponding cross-sections. The CT images acquired using the pulmonary window setting (Figure 3B, Figure 4B, Figure 5B, Figure 6B, Figure 7B, Figure 8B, Figure 9B, Figure 10B, Figure 11B and Figure 12B,C) provided excellent visualisation of various bones and their associated muscles, which were also seen with the soft tissue and bone window settings. Consequently, we observed different bony structures, such as the thoracic vertebrae, ribs, sternum, humerus, femur and pubis, as well as various muscles associated with these bony structures, including the scapulohumeralis, the scapulohumeral caudal, the longissimus colli, the longissimus dorsi, the pectoral, the supracoracoid and the intercostal muscles. In addition, the essential components of the respiratory tract, such as the trachea, the syrinx, the tracheal bifurcation, the course of the right and left main bronchi, the pulmonary vessels as well as the pulmonary parenchyma, could also be distinguished due to their higher attenuation (Figure 3B, Figure 4B, Figure 5B, Figure 6B, Figure 7B, Figure 12B and Figure 12C). Interestingly, the lung parenchyma was well defined, showing a honeycomb-like pattern in all planes, exhibiting an irregular aspect in all puffins (Figure 3B, Figure 4B, Figure 5B, Figure 6B, Figure 7B). Moreover, the right and left primary bronchi were visualised entering into the ventromedial aspect of the right and left lungs (Figure 4B, Figure 5B and Figure 12C). Furthermore, other intrathoracic structures, including the heart and large vessels, such as the two brachiocephalic trunks, were well visualised (Figure 4B and Figure 5B). In addition, the walls of the clavicular, thoracic and abdominal air sacs could be distinguished (Figure 3B, Figure 4B, Figure 5B, Figure 6B, Figure 7B, Figure 8B, Figure 9B, Figure 10B). Concerning the digestive structures, this specific CT window effectively delineated the oesophagus, the right and left hepatic lobe, the glandular and the muscular parts of the puffin stomach, different intestinal segments and the cloaca (Figure 5B, Figure 6B, Figure 7B, Figure 8B, Figure 9B, Figure 10B, Figure 11B). In addition, the dorsal CT images were helpful in observing the bifurcation of the caecum in the right and left caecum (Figure 12C).

The CT images obtained using the soft tissue windows (Figure 3C, Figure 4C, Figure 5C, Figure 6C, Figure 7C, Figure 8C, Figure 9C, Figure 10C, Figure 11C) also provided satisfactory identification of the bony and muscle structures that comprise the coelomic cavity. In addition, this CT window allowed the identification of the oesophagus located ventrally to the vertebral body (Figure 3C, Figure 4C, Figure 5C). Additionally, adequate visualisation of the heart was distinguished with this CT window. Therefore, we could observe the right ventricle surrounded by the right and left hepatic lobe (Figure 5C, Figure 6C, Figure 7C), as well as the main arteries, such as the Right and Left Brachiocephalic Trunks, which appeared with intermediate attenuation (Figure 4C). Moreover, the lumen of the proventriculus and the ventricular folds of the puffin stomach were seen with excellent detail in the left portion of the coelomic cavity (Figure 6C, Figure 7C, Figure 8C, Figure 9C).

The CT images acquired with the use of the bone window setting (Figure 3D, Figure 4D, Figure 5D, Figure 6D, Figure 7D, Figure 8D, Figure 9D, Figure 10D, Figure 11D) demonstrated excellent differentiation between the bones and the soft tissues within the coelomic cavity. In contrast, the heart chambers and the main arteries exhibited an intermediate CT attenuation (Figure 3D, Figure 4D, Figure 5D, Figure 6D, Figure 7D). Additionally, various digestive structures, including the oesophagus, the hepatic lobes, different intestinal sections and the cloaca, as well as the glandular and the muscular part of the puffin stomach, were also observed with intermediate attenuation (Figure 3D, Figure 4D, Figure 5D, Figure 6D, Figure 7D, Figure 8D, Figure 9D, Figure 10D, Figure 11D). Interestingly, this CT window enabled the observation of the left abdominal air sac covering the ventricular surface, which adjoins the left ventral hepatic peritoneal cavity on the left side of the ventriculus (Figure 8D). In addition, the crassus cranioventralis was discernible by its anatomical location ventral to the ventricle, as well as by its round shape and intermediate attenuation (Figure 8D).

**Figure 3 animals-13-02933-f003:**
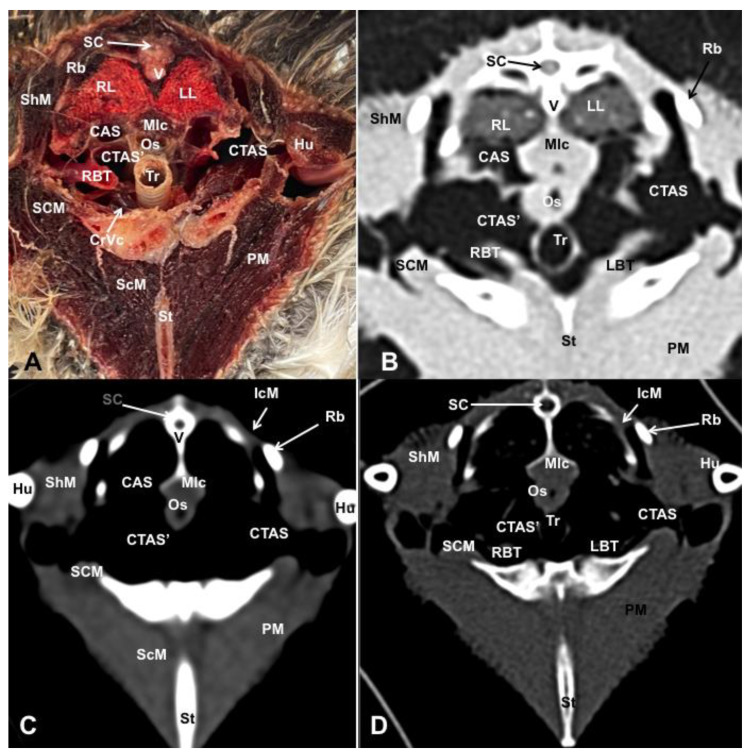
Transverse cross-sectional image (**A**), and pulmonary (**B**), soft tissue (**C**) and bone (**D**) CT images of the coelomic cavity of the *Fratercula arctica* at the level of the trachea corresponding to line I in Figure 1. SC: spinal cord. V: vertebra (vertebral body). ShM: scapulohumeralis muscle. SCM: scapulohumeral caudal muscle. RL: right lung. LL: left lung. Mlc: longissimus colli muscle. Os: oesophagus. Tr: trachea. CTAS: cranial thoracic air sac (left). CTAS’: cranial thoracic air sac (right). CAS: Clavicular Air Sac. LBT: Left Brachiocephalic Trunk. RBT: Right Brachiocephalic Trunk. CrVc: cranial cava vein. St: sternum. PM: Pectoral Muscle (thoracobrachialis muscle). ScM: Supracoracoid Muscle. IcM: intercostal muscle. Hu: humerus. Rb: ribs.

**Figure 4 animals-13-02933-f004:**
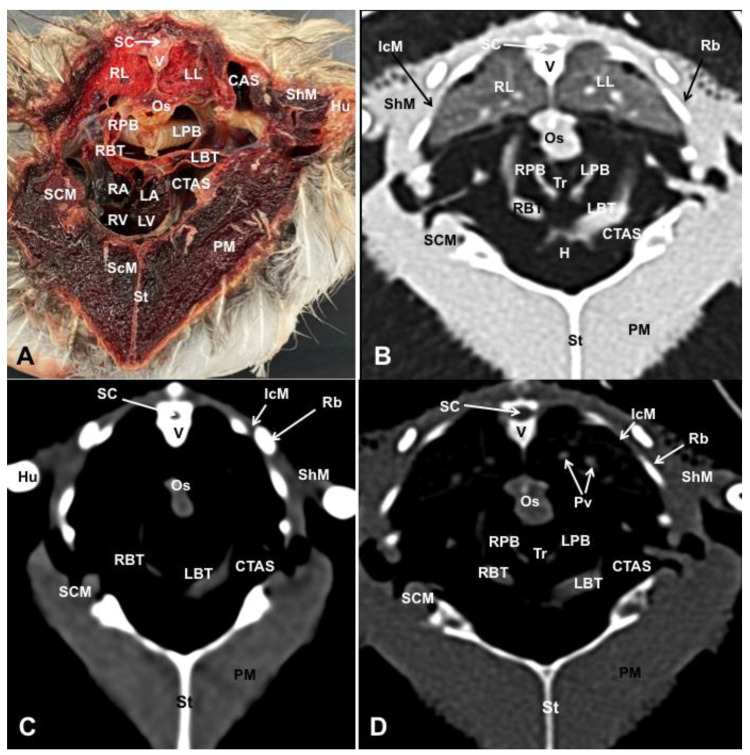
Transverse cross-sectional image (**A**), and pulmonary (**B**), soft tissue (**C**) and bone (**D**) CT images of the coelomic cavity of the *Fratercula arctica* at the level of the heart corresponding to line II in Figure 1. SC: spinal cord. V: vertebra (vertebral body). ShM: scapulohumeralis muscle. SCM: scapulohumeral caudal muscle. RL: right lung. LL: left lung. Pv: pulmonary vessels. Os: oesophagus. Tr: trachea. LPB: Left Primary Bronchus. RPB: Right Primary Bronchus. CTAS: cranial thoracic air sac (left). CAS: Clavicular Air Sac. LBT: Left Brachiocephalic Trunk. RBT: Right Brachiocephalic Trunk. H: heart. RA: Right Atrium. LA: Left Atrium. RV: right ventricle. LV: Left Ventricle. St: sternum. PM: Pectoral Muscle (thoracobrachialis muscle). ScM: Supracoracoid Muscle. IcM: intercostal muscle. Hu: humerus. Rb: ribs.

**Figure 5 animals-13-02933-f005:**
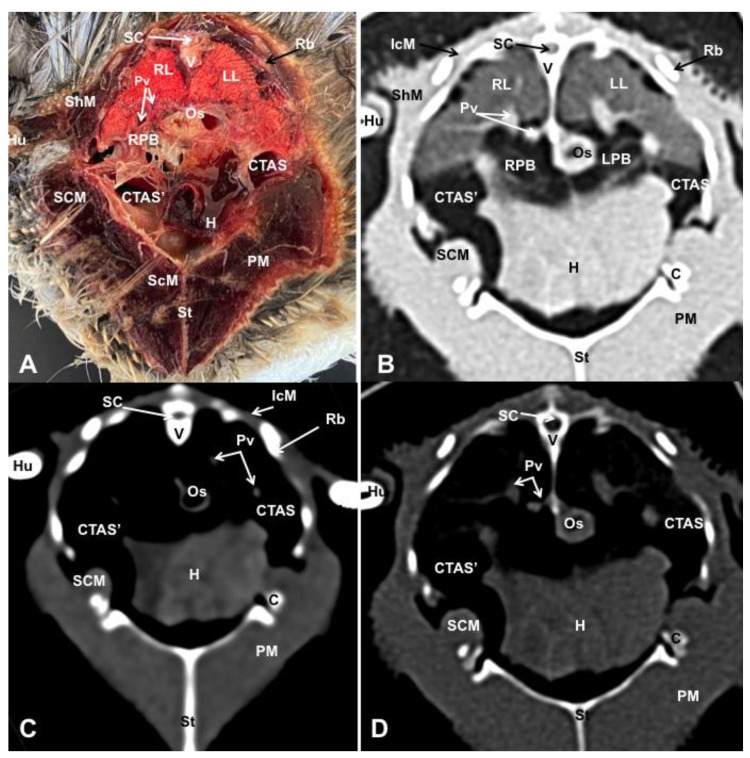
Transverse cross-sectional image (**A**), and pulmonary (**B**), soft tissue (**C**) and bone (**D**) CT images of the coelomic cavity of the *Fratercula* arctica at the level of the cranial thoracic air sacs corresponding to line III in Figure 1. SC: spinal cord. V: vertebra (vertebral body). ShM: scapulohumeralis muscle. SCM: scapulohumeral caudal muscle. RL: right lung. LL: left lung. Pv: pulmonary vessels. Os: oesophagus. LPB: Left Primary Bronchus. RPB: Right Primary Bronchus. CTAS’: cranial thoracic air sac (right). CTAS: cranial thoracic air sac (left). H: heart. St: sternum. C: coracoid bone. PM: Pectoral Muscle. ScM: Supracoracoid Muscle. IcM: intercostal muscle. Hu: humerus. Rb: ribs.

**Figure 6 animals-13-02933-f006:**
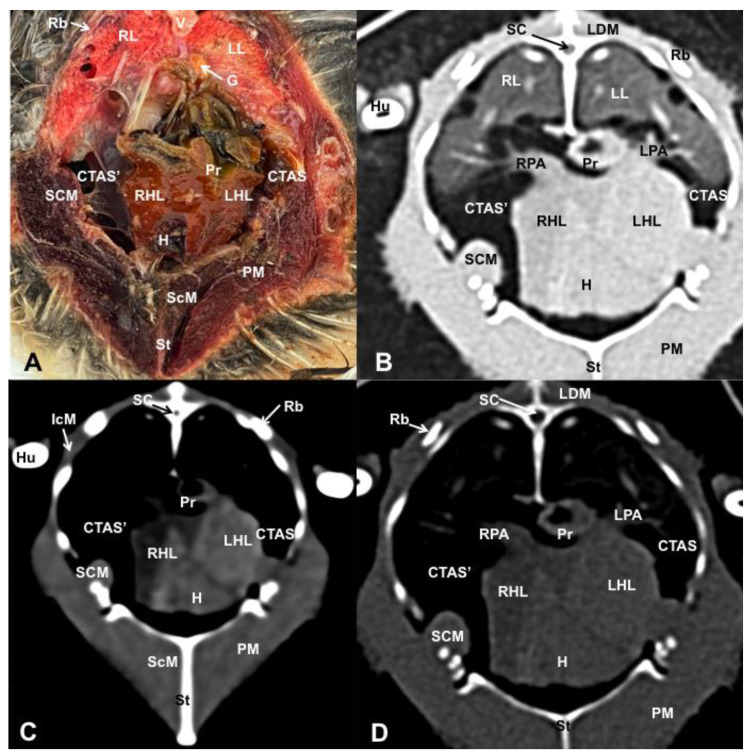
Transverse cross-sectional image (**A**), and pulmonary (**B**), soft tissue (**C**) and bone (**D**) CT images of the coelomic cavity of the *Fratercula arctica* at the level of the liver (hepatic lobes) corresponding to line IV in Figure 1. SC: spinal cord. V: vertebra (vertebral body). LDM: Longissimus Dorsi Muscle. RL: right lung. LL: left lung. RPA: right pulmonary artery. LPA: left pulmonary artery. G: gonad. CTAS: cranial thoracic air sac (left). CTAS’: cranial thoracic air sac (right). Pr: proventriculus. RHL: right hepatic lobe. LHL: left hepatic lobe. H: heart (ventricle). SCM: scapulohumeral caudal muscle. St: sternum. ScM: Supracoracoid Muscle. PM: Pectoral Muscle. IcM: intercostal muscle. Hu: humerus. Rb: ribs.

**Figure 7 animals-13-02933-f007:**
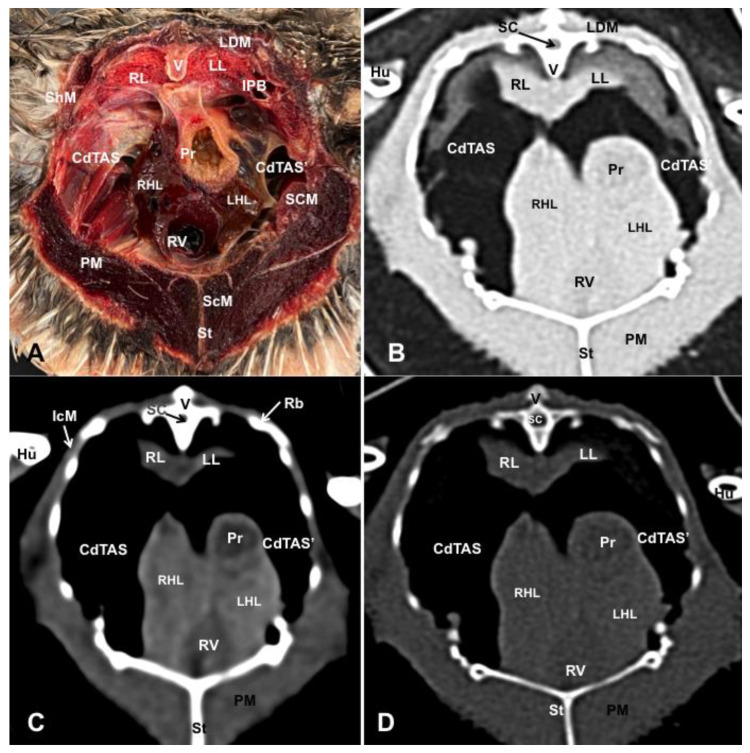
Transverse cross-sectional image (**A**), and pulmonary (**B**), soft tissue (**C**) and bone (**D**) CT images of the coelomic cavity of the *Fratercula arctica* at the level of the caudal thoracic air sac corresponding to line V in Figure 1. SC: spinal cord. V: vertebra (vertebral body and spinal process). LDM: Longissimus Dorsi Muscle. ShM: scapulohumeral muscle. RL: right lung. LL: left lung. CdTAS: caudal thoracic air sac (right). CdTAS’: caudal thoracic air sac (left). Pr: proventriculus. RHL: right hepatic lobe. LHL: left hepatic lobe. RV: right ventricle. St: sternum. ScM: Supracoracoid Muscle. PM: Pectoral Muscle. IcM: intercostal muscle. Hu: humerus. Rb: ribs.

**Figure 8 animals-13-02933-f008:**
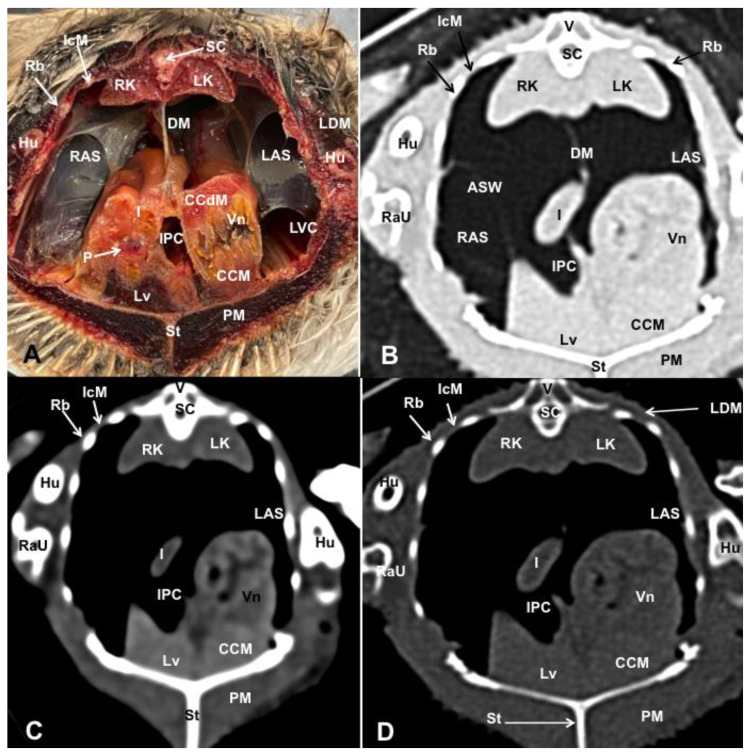
Transverse cross-sectional image (**A**), and pulmonary (**B**), soft tissue (**C**) and bone (**D**) CT images of the coelomic cavity of the *Fratercula arctica* at the level of the kidneys corresponding to line VI in Figure 1. SC: spinal cord. V: vertebra (spinal process). RK: right kidney. LK: left kidney. DM: Dorsal Mesenterium. LAS: left abdominal sac. RAS: right abdominal Sac. ASW: air sac wall. I: intestine. P: pancreas. Vn: ventriculus. CCdM: crassus caudodorsalis muscle. CCM: crassus cranioventralis muscle. IPC: Intestinal Peritoneal Cavity. LVC: left ventral hepatic peritoneal cavity. Lv: liver. St: sternum. PM: Pectoral Muscle. IcM: intercostal muscle. Hu: humerus. RaU: Radius and Ulna. Rb: ribs. LDM: Longissimus Dorsi Muscle.

**Figure 9 animals-13-02933-f009:**
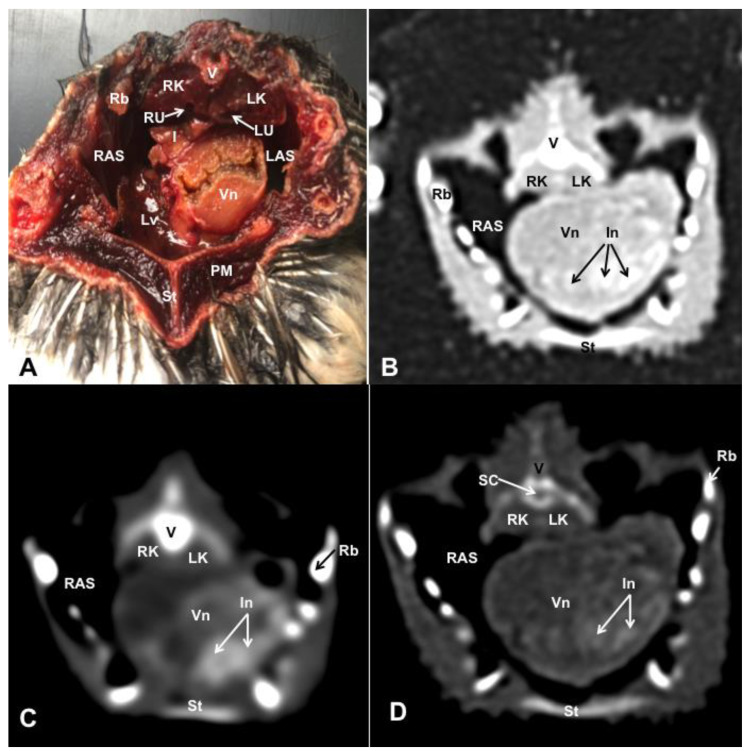
Transverse cross-sectional image (**A**), and pulmonary (**B**), soft tissue (**C**) and bone (**D**) CT images of the coelomic cavity of the *Fratercula arctica* at the level of the ventriculus corresponding to line VII in Figure 1. SC: spinal cord. V: vertebra (vertebral body and spinal process). RK: right kidney. LK: left kidney. RU: right ureter. LU: left ureter. Lv: liver. RAS: right abdominal air sac. LAS: left abdominal air sac. I: intestine. Vn: ventriculus. In: ingesta. St: sternum. PM: Pectoral Muscle (thoracic part). Rb: ribs.

**Figure 10 animals-13-02933-f010:**
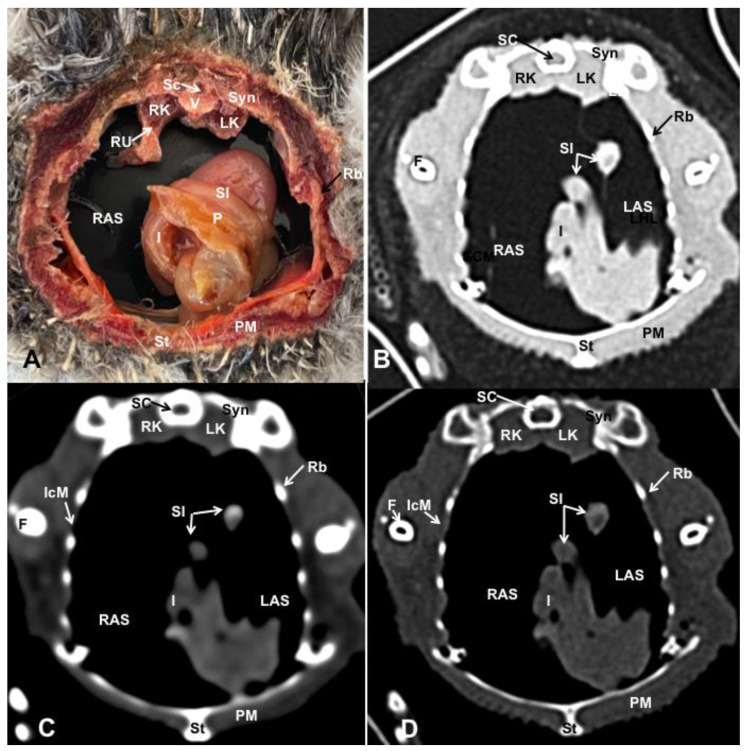
Transverse cross-sectional image (**A**), and pulmonary (**B**), soft tissue (**C**) and bone (**D**) CT images of the coelomic cavity of the *Fratercula arctica* at the level of the caudal end of the right and left kidneys corresponding to line VIII in Figure 1. SC: spinal cord. V: vertebra (vertebral body). Syn: synsacrum. F: femur. RK: right kidney. LK: left kidney. RU: right ureter. RAS: right abdominal air sac. LAS: left abdominal air sac. P: pancreas. I: intestine. SI: small intestine (duodenum). St: sternum. PM: Pectoral Muscle (thoracic part). IcM: intercostal muscle. Rb: ribs.

**Figure 11 animals-13-02933-f011:**
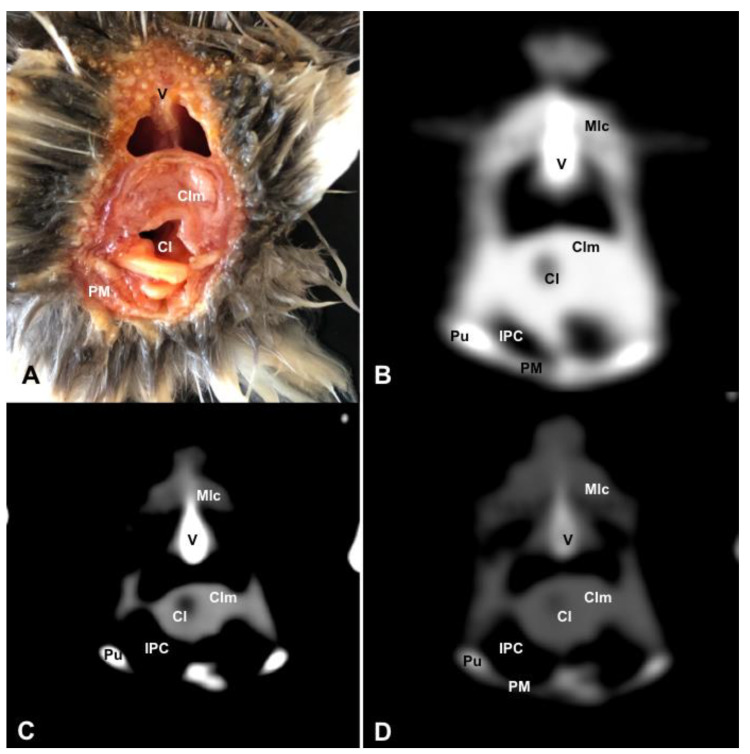
Transverse cross-sectional image (**A**), and pulmonary (**B**), soft tissue (**C**) and bone (**D**) CT images of the coelomic cavity of the *Fratercula arctica* at the level of the cloaca corresponding to line IX in Figure 1. V: vertebra. Clm: cloacal muscles (musculus levator cloacae + musculus transversus cloacae). Cl: cloaca. IPC: Intestinal Peritoneal Cavity. Pu: pubis. PM: Pectoral Muscle (abdominal portion). Mlc: musculus levator caudae.

**Figure 12 animals-13-02933-f012:**
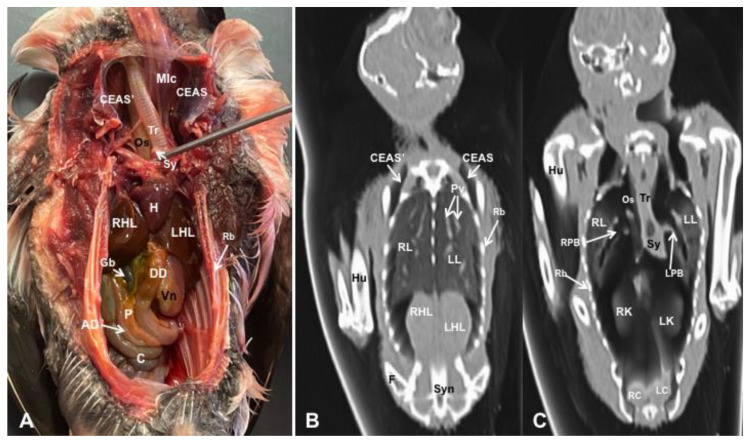
Gross dissection image of the coelomic cavity of the *Fratercula arctica* (**A**), and dorsal MPR volume rendering images in pulmonary CT window of the coelomic cavity at level of hepatic lobes (**B**) and kidneys (**C**). Mlc: longissimus colli muscle. CEAS’: Right Cervical Air Sac. CEAS: Left Cervical Air Sac. Tr: trachea. Sy: syrinx. Os: oesophagus. LL: left lung. RL: right lung. Pv: pulmonary vessels. LPB: Left Primary Bronchus (intrapulmonary portion). RPB: Right Primary Bronchus (intrapulmonary portion). H: heart. LHL: left hepatic lobe. RHL: right hepatic lobe. Gb: gallbladder. DD: Descending Duodenum. AD: Ascending Duodenum. Vn: ventriculus. P: pancreas. C: caecum. RC: right caecum. LC: left caecum. Hu: humerus. F: femur. Syn: synsacrum. Rb: rib.

## 4. Discussion

Birds possess a complex anatomy, very different from mammals, as they lack a distinct separation between the thorax and the abdomen, resulting in the coelomic cavity. Consequently, this cavity has a complex organisation that encloses organs of the circulatory system and the respiratory, digestive and urogenital apparatus [24,25,30,31,32,33,34].

The conventional approach to studying animal anatomy, which involves dissections for visualising organs and internal structures, has undergone a profound transformation with the irruption of modern imaging techniques such as MRI and CT, which are the most advanced diagnostic imaging modalities available for identifying normal anatomy and abnormalities in the internal organs of live animals [9]. In contrast to conventional imaging methods, such as traditional radiographs or ultrasonography, these modern modalities are essential techniques for enabling better morphologic characterisation of the coelomic cavity by providing views of body sections from multiple tomographic planes without repositioning the animal, thereby delivering images with exceptional anatomic resolution in the absence of tissue superimposition, high contrast between different structures, and excellent tissue differentiation that permits the assessment of spatial relationships between organs that are not detected using radiography or ultrasonography. Disadvantages of these procedures include their high cost, the need to sedate and possibly restrain the bird, and the longer examination time in the case of MRI [4,5,6,21,25,26,27]. Due to these shortcomings, the traditional imaging procedures are still the most routinely used because they are fast and low-cost and are widely available in daily avian practice [5,26].

In the present study, coelomic CT images were assessed using pulmonary, soft tissue and bone window settings. The employment of the pulmonary window setting yielded notable benefits in the differentiation of the respiratory tract and diverse vascular structures. As a result, the lung parenchyma, with its honeycomb pattern related to the end-on view of the parabronchi; the course of the two pulmonary arteries supplying each lung; and the paired brachiocephalic trunks were well distinguished with this window setting (Figure 4B, Figure 5B, Figure 6B). Moreover, this CT window and the different planes were quite effective in visualising the extrapulmonary and intrapulmonary portions of the primary bronchus into each lung (Figure 12C). Comparable findings have been reported in previous CT investigations performed on other species, such as neonatal foals [35], reptiles [36] and different avian species [25,26,27]. Meanwhile, the application of the soft tissue window provided additional insights into puffin cardiac topography and its associated blood vessels (Figure 4C, Figure 5C, Figure 6C, Figure 7C). However, the combination of the different CT window settings was not helpful for the visualisation of the four cardiac chambers, which was due to the absence of contrast media administration. Although there are limited studies on the use of contrast media in birds [25,26,27], its application has proven to enhance and facilitate the visualisation of cardiovascular structures and air sacs in live specimens [25,26]. Despite the constraint posed in our study by the use of cadaveric specimens, precluding the administration of intravenous contrast medium, the pulmonary CT window facilitated visualisation of the wall of the thoracic and abdominal air sacs (Figure 7B, Figure 8B and Figure 10B). Nevertheless, studies performed on other bird species described how the air sac walls are not precisely distinguished in CT or radiographs [25,26]. Similar to previous studies [25], the combination of the aforementioned pulmonary CT window with the bone window setting revealed a distinct visualisation of the ventriculus of the stomach, whose content exhibited an amalgamation of air and dense materials (Figure 8C,D and Figure 9C,D). In addition, this particular CT window was quite helpful in distinguishing the muscular layer of the puffin ventriculus, and the relation between the bird heart and the right and left hepatic lobes (Figure 7C,D). In contrast to other avian species, such as parrots [25], whose right lobe is generally larger than the left, the puffin liver presented lobes of similar size. In addition, they showed the gallbladder, which was only distinguishable in the dissected images. This fact could be due to the lack of contrast administration that could impede the visualisation of other organs, including the spleen, the ureters or the pancreas.

In avian species, the sexes are separated, consisting of either male or female genital organs. Nevertheless, several avian species consequently do not show phenotypic sexual dimorphism, making it necessary to employ endoscopy or DNA testing for definitive sex determination [32]. In this study, we identified a single round structure adjacent to the caudal margin of the left lung that closely resembled the left ovary. However, the small size of the puffins suggested that they were juvenile specimens and, therefore, sexually immature animals, which could explain the challenge encountered in distinguishing the gonads in these animals. As stated in other reports described in birds [25], the gonads are visible only in sexually mature individuals of large species.

This CT-based anatomical study has confirmed the suitability of cadavers for evaluating diverse anatomical patterns. However, it is essential to consider the absence of blood flow in dead animals when comparing the results with those obtained from live specimens using contrast media [6,17,18,25]. Results from the present study demonstrated that the utilisation of frozen anatomical sections and dissections was beneficial in identifying various coelomic structures observed on transverse CT images and ensured accurate correlations.

## 5. Conclusions

The current investigation elucidates the spatial arrangement of the different organs within the coelomic cavity of puffins, achieved through the integration of diverse CT window settings in conjunction with anatomical cross-sections. The outcomes underscore the efficacy of CT imaging for the comprehensive study of avian anatomical structure, obtaining high-quality images that could serve as a reference to evaluate Alcidae birds with pathologies at the coelom level. Notably, it is imperative to emphasise that the use of CT as a diagnostic modality in live birds requires the administration of anaesthesia and carries potential complications and mortality risks. Consequently, the utilisation of this technique in live avians demands judicious consideration and rigorous justification by the attending clinician.

## Figures and Tables

**Figure 1 animals-13-02933-f001:**
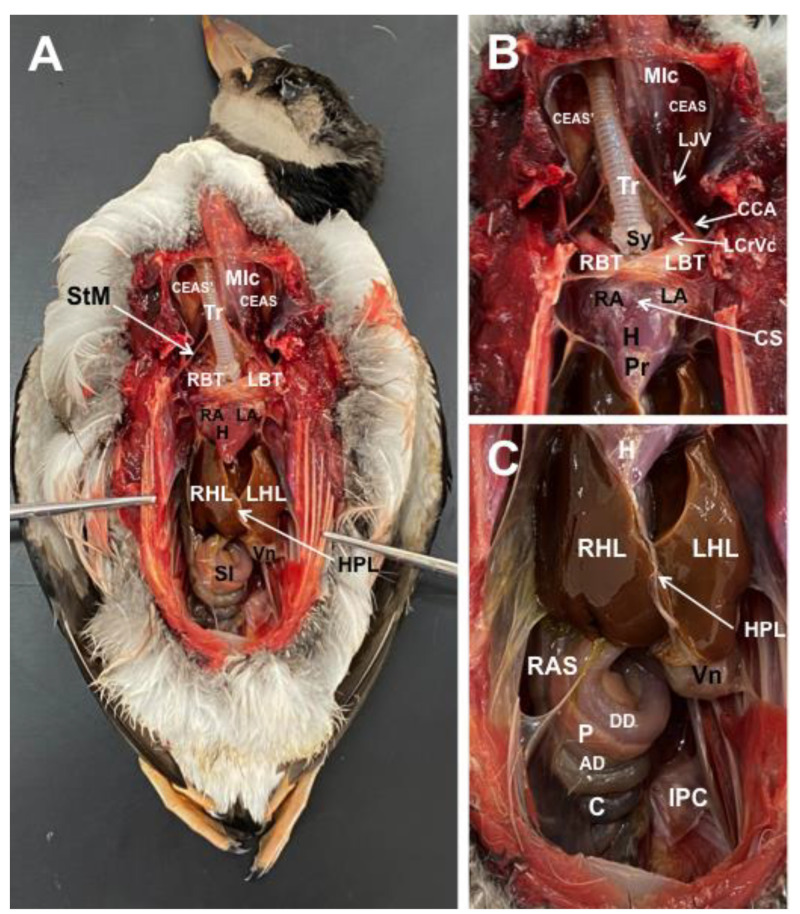
Gross dissection image of the coelomic cavity of the *Fratercula arctica* (**A**), and images of the cardiovascular (**B**) and digestive (**C**) structures. StM: sternotracheal muscle. Mlc: longissimus colli muscle. Tr: trachea. Sy: syrinx. CEAS’: Right Cervical Air Sac. CEAS: Left Cervical Air Sac. LBT: Left Brachiocephalic Trunk. RBT: Right Brachiocephalic Trunk. LJV: left jugular vein. CCA: common carotid artery. LCrVc: left cranial vena cava. H: heart. RA: Right Atrium. LA: Left Atrium. Pr: pericardium. CS: coronary sinus. LHL: left hepatic lobe. RHL: right hepatic lobe. HPL: Hepatopericardial Ligament. DD: Descending Duodenum. AD: Ascending Duodenum. P: pancreas. C: caecum. RAS: right abdominal air sac. Vn: ventriculus. SI: small intestine. IPC: Intestinal Peritoneal Cavity.

**Figure 2 animals-13-02933-f002:**
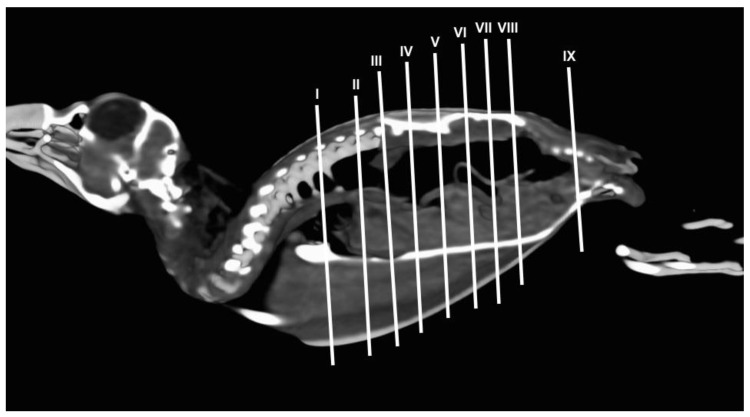
Sagital MPR volume rendering image of the body of an Atlantic puffin. The lines and numbers (I–IX) represent the approximate level of the following transverse cross-sections and CT images.

## Data Availability

The information is available at “https://accedacris.ulpgc.es”, accessed on 25 June 2023.

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
