# Peer review of "Cross-Sectional Anatomy and Computed Tomography of the Coelomic Cavity in Juvenile Atlantic Puffins (Aves, Alcidae, Fratercula arctica)"

_animals, 2023, doi:10.3390/ani13182933_

Round 1

Reviewer 1 Report

To have additional knowledge of the anatomy and pathophysiology of the Atlantic Puffin, Alcidae Fratercula arctica (the species of interest of the present paper), is always valuable. These kind of studies are important for anatomists, conservationists and clinicians. I would like to highlight the use of modern imaging techniques as a complement to more traditional radiological techniques. Also noteworthy is the high quality of the images and the originality of some of the data provided, such as the distribution of the air bags. Among the points to be improved, the addition of an image comparing CT and MR should be interesting, as well as the Discussion approach. In the present form,  the Discussion section is too general. For example, there are no references to the figures, neither to other similar manuscripts, nor what the contribution of the imaging methods used in comparison to traditional radiographs is. In summary, the paper is very valuable but the Discussion should have a better approach. 

The main contribution is the species of interest, as there is practically no information on the Atlantic Puffin (Fratercula arctica).

In my opinion, the authors shouldn't consider specific improvements regarding the methodology. 

The work has an important anatomical basis, so in my opinion nothing more is needed.

The conclusions are consistent with the evidence and arguments presented.

The references are appropriate.

The figures are excellent. Maybe I should combine CT and MR. But the images are very good.

In summary,  I think that the study has enough quality and, in general, fulfills the requirements of Animals' standards.

The manuscript should be reviwed by a native, as there are sentences/paragraphs/expressions that should be changed.

Author Response

Dear Reviewer,

We greatly appreciate your suggestions and comments because we believe they have served to improve the quality of our manuscript. Therefore, we have redone some points of the discussion section, adding more specific information. 

Following your suggestion, we have tried to include the addition of an image comparing CT and MRI. Unfortunately, despite the use of 1.5 T equipment, the size of the animals and the use of carcasses did not allow us to get adequate MR image resolution compared to CT ones.

Reviewer 2 Report

The authors describe the gross anatomy of the coelomic organs in atlantic puffin. The manuscript is interesting, well written and with numerous and useful figures. All the descriptions seems a catalog image atlas. This can be useful for veterinary clinicians that work in zoos and natural parks. 

There are only minor points that can be addressed to improve the manuscript:

The authors generally prise their study on atlantic puffin as an example of avian anatomy for all species. I would suggest being less enthusiastic because there are many bird species vary different in anatomy (for example the coecum is double in numerous species).

The authors should explain why the weight of their puffins had an average mass of 0.251 Kg (with a range of 0.185 - 0.300 Kg) when textbook on this species describe a range of 0.400 - 0.600 Kg).  If, as they affirm, the birds were juvenile, they can add "juvenile" in the title. 

An oversight that is a grave mistake:

page 1, line 26:  Atlantic puffins (Fratercula artica)  instead of   "Fratercula Arctica"

  After a revision of these minor points, I think that your work can be published. 

Author Response

Dear Reviewer,

We sincerely appreciate your comments because they have improved the quality of our manuscript. Therefore, we have addressed the specific minor points.

Thus, In the conclusion section, taking into consideration your comment "There are many bird species that vary differently in anatomy", we have specified that these images could serve as a reference to evaluate Alcidae birds.

To explain better the other point that you suggested, and following your recommendation, we have included the word "juvenile" in the title, and in the materials and methods section.

In addition, we have corrected the mistake of page 1, line 26, as well as in page 2, line 66.